# Genetics and Genomics Teaching in Nursing Programs in a Latin American Country

**DOI:** 10.3390/jpm12071128

**Published:** 2022-07-12

**Authors:** Luís Carlos Lopes-Júnior, Emiliana Bomfim, Milena Flória-Santos

**Affiliations:** 1Health Sciences Center, Nursing Department, Federal University of Espírito Santo (UFES), Av. Marechal Campos, 1468, Maruípe, Vitoria 29043-900, ES, Brazil; 2McGill University, Montreal, QC H3A 0G4, Canada; emiliana.bomfim@mcgill.ca; 3Department of Maternal-Infant Nursing and Public Health Nursing, University of São Paulo at Ribeirão Preto College of Nursing, 3900 Avenida Bandeirantes, Ribeirão Preto 14040-902, SP, Brazil

**Keywords:** competency, curriculum, genetics, genomics, nursing education, teaching

## Abstract

Although the importance of genetics and genomics in nursing education has been widely recognized, surveys carried out in several countries show that these subjects are still limited in nursing undergraduate programs. In Latin America, the teaching of genetics and genomics in nursing programs has never been previously documented. Considering this scenario, we aimed to investigate how genetics and genomics have been taught in undergraduate nursing programs in Brazil. A total of 138 undergraduate nursing program coordinators and 49 faculty members were recruited to participate in this cross-sectional study. After IRB approval, data were collected using an online survey, covering curriculum design, faculty credentials, genetics and/or genomics teaching, as well as their impressions regarding the document “Essential Nursing Competencies and Curricula Guidelines for Genetics and Genomics”. Genetics is taught in most of the investigated courses (67.3%), mainly by biologists (77.6%), with master’s degree (83.7%), and with the syllabus mainly focused on molecular biology. More instructors agreed with Competency 2 (C2) which refers to advocating for clients’ access to desired genetic/genomic services and/or resources including support groups as well as C23 which refer to using health promotion/disease prevention practices that incorporate knowledge of genetic and genomic risk factors, than coordinators. That is, the participants’ type of appointment (instructors vs. coordinators) had a significant effect on their agreement level with competencies C2 (χ^2^ = 6.23, *p* = 0.041) and C23 (χ^2^ = 9.36, *p* = 0.007). Overall, a higher number of participants with both master’s and Ph.D. degrees significantly agreed with competencies C2, C4, which refer to incorporating genetic and genomic technologies and information into registered nurse practice, and C5—demonstrating in practice the importance of tailoring genetic and genomic information and services to clients based on their culture, religion, knowledge level, literacy, and preferred language, when compared to those with Ph.D. only, and those with a master’s degree only (χ^2^ = 8.73, *p* = 0.033; χ^2^ = 8.61, *p* = 0.033; χ^2^ = 8.61, *p* = 0.033, respectively). Our results support reflections on ways to prepare the nursing workforce to deliver personalized nursing care. Additionally, they can be an aid in establishing guidelines for the undergraduate nursing curricula in Brazil and in other Portuguese-speaking countries, as well as in Latin America.

## 1. Introduction

The availability for genomic applications in clinical practice keep transforming healthcare delivery with the potential to increase quality and safety, decrease costs, as well as improve health outcomes [1,2,3,4]. Advances in genomic technologies bring the promise of adapting healthcare through the identification, implementation, and management of genomic risk, along with new treatments [5,6,7]. Genetics and genomics competencies play pivotal roles in nursing practice, including: (i) delivering education for patients and families; (ii) counselling for disorders prevention and health promotion; (iii) collecting the family history; (iv) collaborating with the healthcare team for genetic referrals. These pivotal roles demonstrate that all Registered Nurses (RNs) independently of their speciality or clinical settings might incorporate genetics and genomics knowledge into practice [2,8].

One of the main challenges of nurses’ education in this century is the integration of genetics and genomics into healthcare as well as to deliver nursing care in the Personalized Medicine (PM) Era [1,9,10,11,12,13,14,15,16]. Personalized Medicine aims to customize treatment according to the biological characteristics of individuals or population subgroups [17,18,19]. Thus, the PM Era promises to offer, based on the identification of the patient’s genetic characteristics, the precise medicine in the exact dose and at the right time, thus making it more efficient and reducing the costs of medical care. When considering health care, the debate around PM generally includes a broader conception, i.e., the P4 medicine adds a preventive, predictive, and participatory dimension to personalization [17,19]. The successful implementation of personalized nursing care in the PM Era requires interprofessional collaboration, community outreach efforts, and coordination of care [13].

Despite the surge of interest and attention to precision health, most nurses are not well-versed in precision health or able to understand its implications for the nursing profession. In this sense, it is imperative for the nursing profession to make strategic plans that promote nursing care in the PM Era comprising nursing research, education, clinical practice, and health policy [20].

A debate is needed among undergraduate nursing faculty members on how to include, for instance, new omics approaches in the education of these professionals [10,13,16,20,21]. In this context, further studies are needed to support the teaching plan of nursing courses with regard to genetics and genomics contents, practice, and approach [16,21]. In Latin American countries, little is known on how well essential genetics/genomics content and competences are taught in nursing schools. Therefore, we aimed to investigate how genetics and genomics have been taught in undergraduate nursing programs in Brazil. This study will help educators to establish whether and how nursing curricula is meeting all the essential nursing competencies and curricula expectations in genetics and genomics. It will also shed some light on the shortcomings and pitfalls that need to be addressed in the educational system so that nursing graduates are well prepared to deliver the nursing care in the Personalized Medicine Era.

## 2. Background

Once the Human Genome Project (HGP) was concluded, the National Human Genome Research Institute (NHGRI), National Cancer Institute (NCI), and National Institutes of Health (NIH) planned a genomics education proposal for nursing [8]. The final document, entitled “Essential Nursing Competencies and Curricula Guidelines for Genetics and Genomics”, was concluded after a world effort consensus among researchers and educators [8]. Nowadays, a growing number of activities and international organizations have established or are identifying competencies in genetics and genomics for health professionals [5]. Most health professionals already have competencies in genetics and genomics that are assessed through certification [21].

In a survey conducted in 2005 in North American nursing schools, it was found that 29% of the schools did not address genetics topics in their curricula [22]. A systematic review indicated that the knowledge of genetics and genomics and nurses’ competency in genetics is quite poor [23] and that, based on that evidence, researchers have claimed that there is a deficit of genetic content in many nursing programs [23].

On the other hand, nursing professionals are at the frontline of care in clinical practice, empowering them to serve as a link between families and the health system [1]. In the genomics era, one of the movements to prepare the nursing workforce is teaching based on competencies in genetics and genomics [1,24,25].

Nevertheless, the incorporation of genetics and genomics into clinical practice seems quite inconsistent worldwide [1,26]. In this sense, achieving genomic competency by nursing professionals will not be possible without extensive capacity building amongst nursing faculty members and through continued education of the nursing workforce [1].

Several factors can be attributed to this lack of progress in integrating genetics and genomics into nursing education, e.g., the failure to recognize the value of genetics and genomics in the nursing practice; and a shortage of nursing faculty members prepared to teach these subjects [23]. It is, therefore, imperative to integrate genetics and genomics into the undergraduate curricula by including more genetics and genomics knowledge in the nursing curricula, but mainly, more clinical experience and opportunities to develop genetics and genomics competencies [24,27,28].

Notably, genetics and genomics teaching practice must be guided by a curriculum that is tailored to the national healthcare context and the educational system in which the nursing staff is inserted [29]. In Latin America, especially in Brazil, genetics teaching practices in the nursing curricula have never been explored or documented before. Although studies on the same topic and with similar approaches have been conducted in other countries, such as the US [22], the United Kingdom [30], Japan [31], Taiwan [32], Turkey [28], Singapore [27], and Jordan [33], this is the first nationwide survey conducted in a Latin American country to assess the genetics and genomics teaching practices in undergraduate nursing degrees.

## 3. Methods

### 3.1. Design

A cross-sectional survey design was carried out.

### 3.2. Ethical Consideration

Ethical approval was obtained from the University of São Paulo Institutional Review Board (Reference Number: 1177.2010). Written informed consent was obtained from all participants before data collection.

### 3.3. Participants

The participants were nursing program coordinators at higher education institutions (HEIs) in Brazil, and the faculty that teaches genetics and/or genomics in nursing programs. Based on a survey available on the website of the Brazilian Ministry of Education, we identified 871 HEIs, which comprised the initial population of this study. By the end of the investigation, 138 nursing program coordinators and 49 genetics and/or genomics instructors were selected to take part in and complete this study.

### 3.4. Measures and Outcome Variables

Instruments used in similar studies in Brazil [34] and abroad [35] supported the preparation of our data collection questionnaire, the final version of which was validated through a face and content validity process. It is entitled “Questionnaire on genetics and genomics teaching in undergraduate nursing programs in Brazil”, and consists of four sections, with specific questions regarding: (A) institutions’ characterization; (B) teaching of genetics and genomics, with sociodemographic variables and variables about the education and work carried out by the HEI; (C) courses of genetics and genomics taught by the instructors; (D) the competencies of the North American framework used for this research, called “Essential Nursing Competencies and Curricula Guidelines for Genetics and Genomics”. This framework is divided in two domains: (I) professional responsibilities and (II) professional practice. The North American document was translated into Brazilian Portuguese through a rigorous back translation process after obtaining the approval of the authors. To enable application of the instrument to the Brazilian population, it was culturally adapted and validated using the Delphi technique [36]. After that, questions were designed based on a six-point Likert-type scale (1—Totally agree; 2—Agree; 3—Probably agree; 4—Probably disagree; 5—Disagree; 6—Totally disagree) for each competency.

### 3.5. Pilot Study

Prior to data collection, we performed a pilot study to identify the feasibility of the study design, and the acceptability of interviewees since this topic has not been explored in any previous study. The pilot study helped to reveal difficulties of the participants when answering the items and to estimate the time spent to complete the survey. For data collection an online electronic survey via SurveyMonkey^®^, London, United Kingdom (surveymonkey.co.uk) was used. In total, 20 professionals (15 nursing program coordinators and 5 genetics and genomics instructors) from the HEIs participated in filling out the questionnaire.

### 3.6. Sample Size

The results of the pilot study were used to calculate the sample size for this research. Based on the size of the initially studied population (871 HEIs), the calculation considered the prevalence of 50%, (nursing programs that include the genetics courses) in each unit of analysis (HEI), with α set at 5% and assuming a 20% sample loss. We conducted a simple random draw using the worksheet containing the 871 HEIs with PASS^®^ to obtain a representative sample of the population. Thus, a reliable sample of the 334 HEIs was generated. Of these HEIs, 147 were in the southeast region, 83 in the northeast region, 49 in the south region, 32 in the centre west region, and 23 in the north region of Brazil. We had 23 dropouts; therefore, the final sample size was composed of 311 HEIs (Figure 1).

### 3.7. Data Collection

Data were collected considering the final sample of 311 HEIs, following the same procedures used in the pilot study. In the 311 institutions, only 138 nursing program coordinators and 49 instructors who taught genetics agreed to participate voluntarily in the survey. Firstly, we sent an invitation to the coordinators of the undergraduate nursing programs in Brazil by email. Subsequently, the coordinators were the ones who informed us about the contact of the instructors/professors of the institution that teaches the contents of genetics and genomics to nurses. It is noteworthy that several attempts were made by telephone, in addition to the e-mails, to each instructor/professor, on different days and at different times (morning, afternoon, and evening), to achieve success. Even so, the response rate of this group of participants was low. As a last attempt to increase the number of responses from them as much as possible, a request was made, to support this research, to the Brazilian Society of Genetics (SBG) and to the Brazilian Society of Medical Genetics (SBGM), which resulted in a few more contacts.

The estimated time to complete the online survey was approximately 45 min.

### 3.8. Data Analysis

The data collected via SurveyMonkey^®^ were stored in the platform and exported to Microsoft Excel^®^ spreadsheets. All descriptive statistics were generated in SPSS v. 17.0. The ordinal and nominal qualitative variables were based on absolute numbers and percentages, by means of frequency distribution (%), while the discrete and continuous quantitative variables were presented as mean, standard deviation (SD), median, minimum value, and maximum value. In addition, Likert-scale data were recoded into a new variable with only three categories: agree (i.e., Likert-items 1, 2, 3), disagree (i.e., Likert-items 4, 5, 6), and refused to answer. To look at relationships between the categorical variables, Pearson’s Chi-square test was employed. When the sampling distribution of the test statistic did not show an approximate Chi-square distribution (i.e., expected frequencies were not greater than 5), Fisher’s exact test was used to overcome this issue.

## 4. Results

### 4.1. Questionnaires for the Undergraduate Nursing Program Coordinators

A total of 311 representatives of the HEIs that composed the convenience sample of this work were invited to participate; of these representatives, 138 (44.4%) coordinators of undergraduate nursing programs answered the online survey, and 80.4% filled out the entire instrument.

Table 1 shows the proportion of Brazilian nursing programs that offer genetics and genomics courses in their curricula and have a department of genetics.

Table 2 shows the five competencies with the highest concordance rate among the coordinators regarding the importance of “Essential Nursing Competencies and Curricula Guidelines for Genetics and Genomics”. The Competency (C) with the highest rate of concordance (63.8%) among the coordinators was C19, followed by C23 (59.5%) and C15 (58%).

### 4.2. Questionnaire for Genetics and Genomics Undergraduate Nursing Programs Instructors

Of the 138 coordinators of undergraduate nursing program who participated in the study, 80 (58%) forwarded the contact information of the instructor responsible for the genetics content. Of the 80 instructors we contacted, 49 (61.2%) answered the questionnaire, and 36 (73.5%) of these instructors completed the survey. Table 3 shows the description of our instructor’s sample.

In Table 4, the genetics and/or genomics course is depicted regarding the organization and curricular structure, with a list of the adopted teaching-learning strategies and teaching methodologies.

On average, the credit hours of the genetics and/or genomics course evaluated in this study was 36 h (SD = 14), as reported in Table 5.

Table 6 shows the six competencies with the highest rates of agreement among the interviewed instructors.

More instructors agreed with competencies C2 and C23 than coordinators, and more coordinators refused to give their opinion on these same competencies. That is, the participants’ type of appointment (instructors vs. coordinators) had a significant effect on whether participants would agree or disagree with competencies C2 (χ^2^ = 6.23, *p* = 0.041) and C23 (χ^2^ = 9.36, *p* = 0.007). Overall, a higher number of participants with both master’s and Ph.D. degrees significantly agreed with competencies C2, C4, and C5, when compared to those with a Ph.D. only, and those with a master’s degree only (χ^2^ = 8.73, *p* = 0.033; χ^2^ = 8.61, *p* = 0.033; χ^2^ = 8.61, *p* = 0.033, respectively).

The pattern of responses or level of agreement between coordinators and instructors was significantly different. Overall, more instructors (77.6%) agreed with C2 than coordinators (57.2%), and more coordinators refused to give their opinion on C2. That is, the type of appointment had a significant effect on whether subjects would agree or disagree with C2 [χ^2^ = 6.23, *p* = 0.041]. In addition, the odds of subjects agreeing with C2 were 0.23 higher if they were instructors, than if they were coordinators. In total, there were 138 coordinators (73.8% of the total subjects in the study), and of these, 79 agreed with C2 (57.2% of the total of coordinators) and 9 disagreed (6.5% of the total of coordinators). Further, there were 49 instructors (26.2% of the total subjects in the study), and of those who were instructors, 38 agreed with C2 (77.6% of the total of instructors) and only one of the instructors disagreed with C2 (2% of the total of instructors). In addition, among the coordinators, 50 refused to give their opinion on C2 (36.2% of the total of coordinators), in contrast to 10 instructors that refused to give their opinion on C2 (20.4% of the total of instructors).

Overall, more instructors (73.5%) agreed with C23 than coordinators (65.9%), and more coordinators refused to give their opinion on C23. That is, the type of appointment had a significant effect on whether subjects would agree or disagree with C23 [χ^2^ = 9.36, *p* = 0.007]. In addition, the odds of subjects agreeing with C23 were 10.32 higher if they were instructors, than if they were coordinators. In total, there were 138 coordinators (73.8% of the total subjects in the study), and of these, 91 agreed with C23 (65.9% of the total of coordinators) and 1 disagreed (0.7% of the total of coordinators). Further, there were 49 instructors (26.2% of the total subjects in the study), and of those who were instructors, 36 agreed with C23 (73.5% of the total of instructors), and 4 of the instructors disagreed with C23 (8.2% of the total of instructors). In addition, among the coordinators, 45 refused to give their opinion on C23 (33.3% of the total of coordinators), in contrast to 9 instructors that refused to give their opinion on C23 (18.4% of the total of instructors).

Additionally, the pattern of responses (i.e., agreed, disagreed, refused to answer) between instructors with different graduate degrees was significantly different. Overall, those with both master’s and Ph.D. degrees (90.9%) agreed more with C2 than those with a Ph.D. only (83.3%) and those with a master’s degree only (61.9%). That is, having one or more graduate degrees had a significant effect on whether subjects would agree or disagree with C2 [χ^2^ = 8.73, *p* = 0.033]. Overall, those with both master’s and Ph.D. degrees (90.9%) agreed more with C4 than those with Ph.D. only (66.7%) and those with master’s degree only (57.1%). That is, having one or more graduate degrees had a significant effect on whether subjects would agree or disagree with C4 [χ^2^ = 8.617, *p* = 0.033]. Additionally, those with both master’s and Ph.D. degrees (86.4%) agreed more with C5 than those with Ph.D. only (66.7%), and those with master’s degree only (57.1%). That is, having one or more than one graduate degree had a significant effect on whether subjects would agree or disagree with C5 [χ^2^ = 8.61, *p* = 0.033].

## 5. Discussion

In this study, our purpose was to investigate how genetics and genomics have been taught in undergraduate nursing programs in Brazil. Despite the contents of genetics and genomics being present in the courses in Brazil, they are not being taught in the way that the international nursing literature indicates that their teaching should be. We also identified some shortcomings in the genetics and genomics teaching of the nursing programs, especially from a clinical standpoint.

Our results also showed that only 17 HEIs (12.3%) have a department of genetics. One of the reasons for this low percentage can be that, in Brazil, as in many other countries, the field of genetics research and education was developed within the area of biology [37,38]. In medical and nursing education, it is crucial that genetics teaching integrates both basic and clinical subjects, with practical applications to the healthcare services context [30]. However, a limiting factor, as pointed out by this study, is that most of the genetics instructors are biologists with no clinical background. In our study, the high percentage of biologists teaching genetics in nursing programs (77.6%) may reflect the historical relationship between genetics and the biological sciences area.

In general, in the healthcare professions, the teaching of genetics and genomics has been focused on the theoretical contents of the molecular and cellular bases of typical and altered organic processes [6,8,10]. However, fundamental concepts of genetics and genomics are limited for clinical situations. Thus, professionals are trained to know nucleic acids, their functioning and techniques for their study, but are unaware of concepts such as: birth defects; understanding the clinical aspects of syndromes, that are frequent in the population; genetic variability; pharmacogenetics/pharmacogenomics; genotype/phenotype correlations; penetrance; expressiveness; epigenetics; omics sciences’ and others necessary for understanding the origin of pathological processes, their degree of heredity, prognosis, as well as therapeutics [6,8,9,10,11].

It is necessary to emphasize the importance of this knowledge for the training of all healthcare professionals, especially nurses. In everyday clinical practice, the nursing professionals is on the front line of care, which allows them to act as an important link between the families served and the other members of the health team [9]. Nurses is uniquely positioned, as they are in direct and constant contact with the subjects under their care, during all phases of the life span, being responsible for relieving their suffering, providing health education, providing patients and their families with greater understanding of the scientific aspects that support the diagnosis, surveillance, and treatment [9,21,24].

These finding is in line with a previous study conducted in Japan that sought to verify the qualification of the genetics instructors of bachelor nursing programs also found a higher prevalence of biologists (over 50%), followed by professor doctors (41.2%) [31]. However, in our study, 8.2% of the faculty members are nurses, which can be a reflection of the advancement of medical and clinical genetics and the influence of the genomics era that has sparked the interest of nurses and other health professionals, who subsequently choose to specialise in this area [9,10,39,40]. Since the teaching of genetics has its origins embedded within the biology-related disciplines and departments, it is important for biologists who teach genetics in a nursing program to recognize that any efforts to enhance their performance does not merely depend on instructor-centred aspects, but mostly on the course format and content, which should incorporate clinical, ethical, and social applications of genetics content [38].

Regarding the titles, most of the instructors held a master’s degree (53%) and a doctorate degree (55.1%) in the areas of genetics, molecular biology, and/or biology. The frequency of specialization programs was lower (14.2%). In general, this reveals that there is a reasonable number of highly specialized instructors in these areas of expertise, but who do not necessarily have the expertise to teach nurses from the perspective of Personalized Nursing Care. The high number of graduate degrees can be explained by the institutional requirements employed by the Brazilian Ministry of Education, which determines the minimum qualification of undergraduate instructors (i.e., often requires a doctoral or master’s degree).

Interestingly, most nursing programs (67.3%) only offer basic genetics and molecular biology under their genetics course. The lack of integration of the disciplines of genetics and genomics with the clinical practice of nurses is a challenge to be overcome in the training of nurses in the Latin America, especially considering the current Personalized Medicine Era. Moreover, 12.2% did not offer the class separately, but rather as a part of other classes, modules, or courses, such as cell biology. A survey conducted in 2005 with North American nursing schools found that 71% of the schools included genetics in their curricula [22]. Another study in Japan found that 66.7% of all undergraduate nursing programs in the country included some content of genetics in the curricula [31]. In a more recent study conducted in Turkey, most participants (81.1%) reported that a genetics course was not included in the curriculum of their undergraduate programs [28].

Indeed, at the core of nursing and medical curricula, genetics teaching focuses on the theory of medical genetics and on molecular and cellular basic principles. Although the course of for healthcare professionals is included in several curriculum matrices, reinforcing the methodological quality and new educational initiatives is important to enhance the teaching-learning process of this subject matter [41].

In Japan, a greater percentage (70%) of molecular content present in the syllabus has been reported (i.e., DNA structure and function, RNA, and gene and chromosome structure). In contrast, only 30% included content such as genetic counseling, oncogenetics, and gene therapy tailored for nursing clinical practice [31]. Contents such as genetic testing, pharmacogenomics, ethical aspects of healthcare based on genetic and genomics and personalized medicine are barely covered [42]. Undoubtedly, the benefit of integrating omics (genomics, transcriptomics, proteomics, metabolomics, epigenomics) into nursing research is well established worldwide [3,43,44,45]. These omics approaches are crucial for elucidating the molecular mechanisms underlying the risk, manifestation, as well as symptoms trajectory of disease, to identify biological signatures associated with several health outcomes [3].

Knowledge of basic terminology in genetics (e.g., patterns of inheritance; gene/environment interaction/behaviour at the onset and in the treatment of diseases; the difference between clinical diagnosis and genetic predisposition), along with the ability to use and record the results of the three-generation family history tool to determine patients with or at risk of genetic disorders, and awareness of the confidential nature of genetic information, are some of the key competencies outlined by international institutions and societies that should be mastered by healthcare professionals [9,21,46].

In a recent Brazilian study with fifty-four nurses and physicians from primary healthcare, most participants (85.2%) stated they were taught genetics contents during college; however, most of them (77.8%) mentioned that they did not feel prepared to deliver genomics-based care in primary care settings, mainly due the lack of knowledge [38].

Mastering the concepts of genetics and genomics is essential for contemporary nursing practice [21,47]. Therefore, the number of studies that evaluate the knowledge of nurses and nursing instructors on genetic content have increased in recent years [48]. A study in the US with a sample of 495 genetics instructors in nursing programs showed that instructor’s knowledge on fundamental concepts of genetics was like the knowledge of their own students, revealing more limitations regarding the basic concepts [49].

The most widely used teaching strategy in genetics observed in our study were traditional-based lectures (81.6%). Studies have sought to identify the preferred teaching-learning strategies and resources of genomics according to professors and students [50]. While books and sites were the most widely used resources for educators in the UK, access to genetic services and case studies were identified as being the most useful forms of learning [51]. A study that investigated the perceived knowledge of 190 nurses from Taiwan regarding genetics and genomics, found that the most effective method for teaching this course was lectures/conferences, with 95.3% of the responses. Other approaches considered as being fairly productive were seminars, reading papers, and case studies (92.1%) [52]. Additionally, in our study, the average credit hours of evaluated genetics and genomics courses were 36 h. Similar studies report that, despite the need to increase the credit hours for the genetics and genomics study, this credit hours have increased in relation to previous programs [32].

With regard to the “Essential Nursing Competencies and Curricula Guidelines for Genetics and Genomics”, most of the coordinators of the Brazilian nursing programs, who were nurses, and the instructors of genetics were probably unaware of its existence.

It is noteworthy that efforts have been made since the 1990s, to better prepare nurses in genetics and genomics by four influential organizations: American Association of Colleges of Nursing (AACN); the American Nurses’ Association; the American Academy of Nursing; and the International Society of Nurses in Genetics, who each published position or competency statements [8,9,10].

The “Essential Nursing Competencies and Curricula Guidelines for Genetics and Genomics” is a useful instrument for guiding the incorporation of genetics and genomics into nursing curricula and practice. The specific genetic/genomic competencies can also be used to guide curriculum assessment and planning, continuing education, and specialty certification, as well as individual competency evaluation. Prior to the publication of the “Essentials”, studies reported limited nursing competency in genetics and genomics [22,35], and a goal of their development was to create measurable indicators of success [8]. As such, the “Essentials” document is useful both nationally and internationally for program evaluation [9,11].

Surprisingly, there was a convergence between the responses of the nursing program coordinators and the genetics instructors since the highest concordance rates were attributed to these two groups for the competencies C15 and C19. These competencies, respectively, refer to identifying clients who can benefit from genetics and genomics information and/or services based on the collected data and facilitating patients access to specialized genetics and genomics services, when necessary [8].

In the multidisciplinary team, nurses occupy a prominent position since they are in direct and constant contact with the individuals under their care [13,14,15,53]. In some areas, such as oncology, genetics and genomics are intrinsically related to nursing care through health education activities, counseling, and pharmacogenomics translation into clinical practice [54,55,56,57,58,59,60,61].

The results of a systematic review indicate that nurses still lack the competence to provide healthcare based on genetics and genomics to clients in several clinical conditions. No studies reported in this review showed that nurses have the appropriate levels of knowledge and/or skills to act according to the core competencies in genetics and genomics established internationally [23]. These core competencies established that nurses must be able to determine whether a patient would benefit from a referral and must be capable of making this referral [30].

A study conducted in Singapore showed that nurses have been using genomics assessments more frequently, but specifically to collect family history and to investigate environmental and physical risk factors, rather than genomics interventions itself [27]. In another study carried out with a sample of nurses in Jordan, it was found that the most of interviewed nurses (86%) had an inappropriate level of knowledge with regard to obtaining family history, and to provide genetic information for affected people or high-risk families. Additionally, the study showed that nurses did not perceive themselves as responsible for important genetic-related tasks, such as “genetic diagnosis” and “explaining the results of genetic disorders” [33]. This perception seems to be like the perception of the Brazilian nursing program coordinators and genetics instructors who participated in this investigation.

Finally, the Global Genomics Nursing Alliance (G2NA) was recently established to accelerate the integration of genomics into clinical practice. The focus of the G2NA is the general nursing community [21]. According to Calzone et al. [21] the primary intent of this genomic knowledge mobilization is to optimize resource accessibility and reduce duplication of efforts through leadership, collaboration and sharing within the nursing international community. The ultimate aim is to increase nursing capacity to integrate genomics into nursing practice through supporting improvements in genomic literacy [21].

It is important to consider three additional critical steps that are yet to be achieved: (i) core genomic nursing competencies must be mapped in academic curricula and aligned with national policies; (ii) standardized tools must be developed to evaluate genomic nursing competencies at basic and advanced levels in academia and in practice, particularly in primary health care; and (iii) accreditation standards for genetic nursing education need to be developed in graduate programs for specialty tracks in genomics [23,62].

Additionally, it is also important to highlight that the content of the undergraduate nursing programs should be presented, emphasizing the role of genetics and genomics for nursing. For instance, the AACN points out that the skills that nursing professionals with a bachelor’s degree need to develop include: (a) receiving training in genetics and genomics sciences, pharmacogenetics, and pharmacogenomics; (b) knowing the social impact of genetic and genomic for health policies as well as stakeholders; (c) accessing protective and predictive factors, considering those of genetic origin, which influence the health of individuals, families, communities, and populations; (d) surveying health history, using the pedigree constructed from information collected from family history, with genetic risks, for current and future health problems; (e) accessing the current knowledge in genetics and genomics, including personalized therapies; and (f) recognizing the relationship of genetics and genomics with health, prevention, screening, diagnosis, prognosis, and treatment [63].

Future nurses need to understand not only the foundations of genetics and genomics, but also the implications of these sciences for their clients including psychosocial issues and data sharing in genomic research, for example [64]. Therefore, Ethical, Legal, and Social Issues (ELSI) in clinical genetics research is quite important as well in this process [65]. Students’ feedback regarding the potential of genetics and genomics in the clinical practice, as well as how to incorporate in experiential resources into the courses, are also needed to be included.

This study must be analysed considering its limitations. Despite numerous attempts to enlarge the number of participants, the sample size was small. In addition, the group of coordinators and faculty members were not balanced, which might have resulted in the difference in the competencies observed. Advances in genomics arena are transforming nursing practice, and the continuous improvement of genetics educational initiatives will support the training of genomically literate nurses. Therefore, our findings can certainly be an aid in establishing guidelines for undergraduate nursing curricula and to plan continuing training and education in Brazil and in other Portuguese-speaking countries, as well as in Latin America.

In summary, our findings provide an important update on how genetics and genomics is taught in Brazilian nursing schools. Additionally, our results suggest that there are still deficiencies in the teaching of genetics and genomics in the curricula of Brazilian nursing programs, especially if considering the essential knowledge elements and practice indicators from the “Essential Competencies”.

Additionally, our results show that clinical contextualization and student clinical practicums in genetic-related contexts are still inexpressive in the nursing curricula. Taken together, these data may help nursing educators in developing effective strategies to redesign genetics and genomics teaching practices for nursing students, thereby increasing their awareness of the pivotal importance of genetics knowledge within their nursing responsibilities in general professional practice. To effectively translate genetics and genomics knowledge into nursing practice, it is essential to incorporate experiential resources into the courses early. These resources must be developed in the context of the healthcare continuum through, for example, objective structured clinical examinations (OSCEs) that can strongly emphasize the practical applications of genetics and genomics in clinical scenarios.

## Figures and Tables

**Figure 1 jpm-12-01128-f001:**
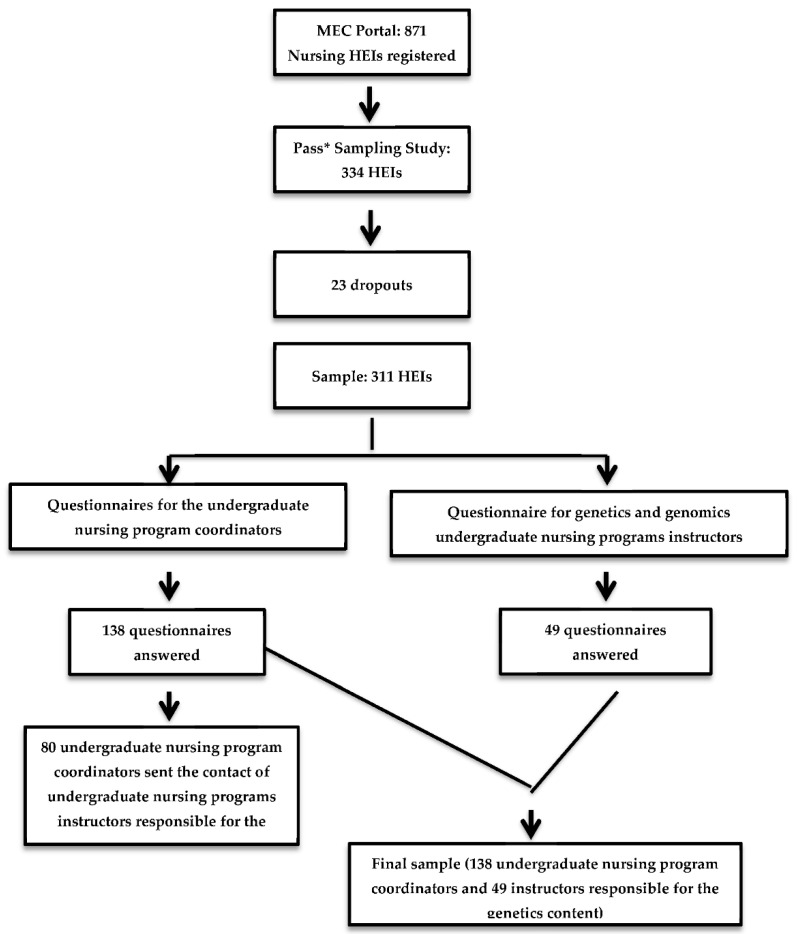
Recruitment flowchart. Abbreviations: MEC, Brazilian Ministry of Education; HEIs, Higher Education Institutions; PASS*, Power Analysis and Sample Size.

**Table 1 jpm-12-01128-t001:** Characterization of the HEIs regarding the existence of instructors and departments of genetics.

**Variables**	**n**	**%**
**HEI has instructor responsible for the genetics content for the undergraduate nursing programs**		
No	15	10.9
Yes	110	79.7
Not informed	13	9.4
**Existence of a genetics department at the HEI**		
No	103	74.7
Yes	17	12.3
Not informed	18	13.0
**Number of genetics instructors at the HEI**		
Only one	65	47.1
Two	25	18.1
Three	10	7.2
Four	5	3.7
Five	2	1.4
Six	2	1.4
Eight	1	0.8
Not informed	13	9.4

Abbreviation: HEI, Higher Education Institution.

**Table 2 jpm-12-01128-t002:** Perceptions of undergraduate nursing program coordinators about the “Essential Nursing Competencies and Curricula Guidelines for Genetics and Genomics”.

Core Competencies of Nursing for Genetics and Genomics	Totally Agree n (%)	Agree n (%)	Probably Agree n (%)	Probably Disagree n (%)	Disagree n (%)	Totally Disagree n (%)
** *PROFESSIONAL RESPONSIBILITIES DOMAIN.* ** *Nurses must have knowledge and skills in genetics and genomics to:*						
** *C4.* ** *Incorporate genetics and genomics technologies and information into nursing practice.*	37 (26.8)	37 (26.8)	16 (11.6)	3 (2.2)	1 (0.7)	0 (0.0)
** *PROFESSIONAL PRACTICE DOMAIN. (a)* ** *By integrating knowledge of genetics and genomics into nursing assessments, nurses can:*						
***C12.****Critically analyze the history and physical assessment findings for genetic, environmental, and genomic influences and risk factors*.	27 (19.6)	35 (25.4)	15 (10.9)	4 (2.9)	5 (3.6)	2 (1.4)
** *PROFESSIONAL PRACTICE DOMAIN. (b)* ** *Integrating knowledge of genetics and genomics into the nursing assessment allows nurses to:*						
***C15.*** *Identify clients who may benefit from specific genetic and genomic information and/or services based on assessment data.*	39 (28.3)	41 (29.7)	17 (12.3)	0 (0.0)	1 (0.7)	0 (0.0)
** *PROFESSIONAL PRACTICE DOMAIN. (c)* ** *Integrating knowledge of genetics and genomics into the nursing assessment enables referrals, because nurses can:*						
***C19.*** *Facilitate referrals for specialized genetic and genomic services for clients as needed.*	45 (32.6)	43 (31.2)	12 (8.7)	3 (2.2)	2 (1.4)	1 (0.7)
** *PROFESSIONAL PRACTICE DOMAIN. (d)* ** *Integrating knowledge of genetics and genomics into the nursing assessment allows the provision of education, care, and support, because nurses can:*						
** *C23.* ** *Use health promotion/disease prevention practices that incorporate knowledge of genetic and genomic risk factors.*	39 (28.3)	43 (31.2)	9 (6.5)	0 (0.0)	0 (0.0)	1 (0.7)

**Table 3 jpm-12-01128-t003:** Demographic data and characterization of the instructors responsible for the genetics content for the undergraduate nursing programs.

Variables	n (%)
**Field of Graduation**	
Biology	38 (77.6)
Biomedicine	1 (2.0)
Medicine	2 (4.1)
Nursing	4 (8.2)
Biochemical Pharmacy	3 (6.1)
Physical Sciences	1 (2.0)
**Year of completion of graduate studies**	
Before the completion of the HGP (i.e., before 2003)	33 (67.3)
After the completion of the HGP (i.e., after 2003)	16 (32.7)
**Type of HEI from which the instructor graduated**	
Pubic	16 (32.7)
Private	27 (55.1)
Not informed	6 (12.2)
**Titration**	
Specialization	
No	30 (61.2)
Yes	19 (38.8)
Area: Genetics and/or Molecular Biology and/or Biology	7 (14.2)
Other areas	12 (24.6)
Master’s Degree	
No	8 (16.3)
Yes	41 (83.7)
Area: Genetics and/or Molecular Biology and/or Biology	26 (53.0)
Other areas	15 (30.7)
Doctorate Degree	
No	18 (36.7)
Yes	31 (63.3)
Area: Genetics and/or Molecular Biology and/or Biology	27 (55.1)
Other areas	4 (8.1)
Post-Doctorate	
No	42 (85.7)
Yes	7 (14.3)
Area: Genetics and/or Molecular Biology and/or Biology	5 (10.2)
Other areas	2 (4.1)
**Has an administrative position at the HEI**	
No	22 (44.9)
Yes	27 (55.1)
Coordinator of undergraduate Nursing Program	1 (2.0)
Coordinator/Head of the Genetics Course	17 (34.7)
Another Coordinating Position	9 (18.4)
**Teaching position at IES**	
Hired	5 (10.2)
Assistant	9 (18.3)
Adjunct	10 (20.5)
Associate	11 (22.5)
Full Professor	1 (2.0)
Visiting Professor	0 (0.0)
Not informed	13 (26.5)
**Teaches genetics content for another Program, besides Nursing**	
No	12 (24.5)
Yes	37 (75.5)
**Teaches genetics content in another HEI**	
No	38 (77.6)
Yes	11 (22.4)
**Also teaches other courses in the same HEI or another HEI**	
No	7 (14.3)
Yes	42 (85.7)
**Is responsible to accompanying practice activities of the course**	
No	3 (6.1)
Yes	13 (26.5)
Not applied	21 (42.9)
Not informed	12 (24.5)
**Member of any National or International Genetics or Genomics Society**	
No	32 (65.3)
Yes	17 (34.7)

Abbreviations: HGP, Human Genome Project; HEI, Higher Education Institution.

**Table 4 jpm-12-01128-t004:** Description of the curriculum organization and teaching-learning strategies of the genetics and/or genomics course.

Variables	n (%)
**The undergraduate nursing program has a specific course for genetics/genomics**	
Yes	33 (67.3)
No, because the HEI uses other forms of curriculum organisation.	4 (8.2)
No, but the content is taught in other courses.	6 (12.2)
Not informed	6 (12.2)
**Type of the discipline**	
Elective or Optional	3 (6.1)
Mandatory	40 (81.6)
Not informed	6 (12.2)
**Name given to the discipline**	
Genetics	13 (26.5)
Genetics and Evolution	5 (10.2)
Human Genetics	11 (22.4)
Clinical/Medical Genetics	2 (4.1)
Cellular and Molecular Biology	1 (2.0)
Cell Biology and Genetics	2 (4.1)
Biology	4 (8.2)
Embryology and Genetics	1 (2.0)
Nursing in Genetics and Genomics	1 (2.0)
Biological Sciences	1 (2.0)
Not informed	8 (16.3)
**Semester in which the course is taught**	
First semester	9 (18.4)
Second semester	18 (36.7)
Third semester	5 (10.2)
Fourth semester	2 (4.1)
Fifth semester	2 (4.1)
Not informed	13 (26.5)
**Type of evaluation used**	
Summative Evaluation	28 (57.1)
Formative Assessment	10 (20.4)
Summative and Formative Assessment	2 (4.1)

Abbreviations: HEI, Higher Education Institution.

**Table 5 jpm-12-01128-t005:** Credit hours of the genetics and/or genomics course and the adopted teaching-learning strategies and teaching methodologies.

Variables	Mean	Median	SD	Min.	Max.
**Credit hours total of the course (hrs.)**	36	50	±14	20	80
**Credit hours of the course dedicated for:**					
Molecular Biology	25	8	±8	2	40
Basic Genetics/Human Genetics	14.6	18	±12	4	40
Genetic tests	2.7	5.5	±6	1.5	15
Genetic services and the role of Nursing	3.1	7.5	±8	2	27
**Genetics/genomics teaching strategies:**					
Theory classes	34	27	±10	4	60
Lectures or conferences	0.5	4	±1	3	15
Seminars	2.5	7	±3	2	20
Discussion of clinical cases	1.8	5.5	±2	2	20
Tutorials	0.15	2	0	2	6
Directed studies	1.4	5	±1	2	7
Bibliographical search	0.8	4	±1	2	6
Distance learning	0.4	4	0	2	14
Videos	0.7	3.5	0	1	6
Problem-Based Learning	0.3	6	0	6	10
Problem questioning	0.5	4	0	3	6
**Practice hours of genetics/genomics teaching:**					
Practice lessons in the classroom	2.4	10	±2	4	30
Practice lessons/activities in the laboratory	2.3	20	±2	3	34
Practical activities at the nursing unit	0.2	3	0	3	8
**Genetics/genomics teaching practices across lifespan:**					
Pediatrics	1.6	15	±1.5	4	27
Preconception period	1.3	6	±1	3	20
Prenatal	1	5	±1	3	8
Neonatal Period	0.7	4	0	2	6
Adolescent	0.4	3	0	3	5
Adult	3.2	10	±10	5	40
Elderly	0.3	3	0	3	4

**Table 6 jpm-12-01128-t006:** Perceptions of the instructor responsible for the genetics content with regard to “Essential Nursing Competencies and Curricula Guidelines for Genetics and Genomics”.

Essential Nursing Competencies for Genetics and Genomics	Totally Agree n (%)	Agree n (%)	Probably Agree n (%)	Probably Disagree n (%)	Disagree n (%)	Totally Disagree n (%)
** *PROFESSIONAL RESPONSIBILITIES DOMAIN. (a)* ** *Nurses must have knowledge and skills in genetics and genomics to:*						
***C1.*** *Recognize when one’s own attitudes and values related to genetic and genomic science may affect care provided to clients.*	22 (44.9)	18 (36.7)	0 (0.0)	0 (0.0)	0 (0.0)	0 (0.0)
***C2.*** *Advocate for clients’ access to desired genetic/genomic services and/or resources including support groups.*	20 (40.8)	12 (24.5)	5 (10.2)	0 (0.0)	2 (4.1)	0 (0.0)
***C4.*** *Incorporate genetic and genomic technologies and information into registered nurse practice*.	21 (42.9)	13 (26.5)	2 (4.1)	3 (6.1)	0 (0.0)	0 (0.0)
***C5.*** *Demonstrate in practice the importance of tailoring genetic and genomic information and services to clients based on their culture, religion, knowledge level, literacy, and preferred language.*	21 (42.9)	13 (26.5)	2 (4.1)	1 (2.0)	1 (2.0)	0 (0.0)
** *PROFESSIONAL PRACTICE DOMAIN. (b)* ** *The integration of knowledge in genetics and genomics into nursing assessment allows nurses to:*						
***C15.*** *Identify clients who may benefit from specific genetic and genomic information and/or services based on assessment data.*	22 (44.9)	14 (28.5)	1 (2.0)	1 (2.0)	1 (2.0)	0 (0.0)
** *DOMINION OF PROFESSIONAL PRACTICE. (c)* ** *The integration of genetics and genomics knowledge into the nursing assessment enables referrals, because nurses can:*						
***C19.*** *Facilitate referrals for specialized genetic and genomic services for clients as needed.*	19 (38.8)	16 (32.7)	4 (8.2)	0 (0.0)	0 (0.0)	0 (0.0)

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
