# Peer review of "Genetics and Genomics Teaching in Nursing Programs in a Latin American Country"

_jpm, 2022, doi:10.3390/jpm12071128_

Round 1

Reviewer 1 Report

In this article, the authors investigate how genetics and genomics are taught in the undergraduate nursing programs in Brazil. They have developed a thorough questionnaire and include both instructors and coordinators of the courses from higher education institutions in Brazil to fill out the survey. They use apt statistical methods to analyze and deduce correlations. The authors also compare their study and conclusions with other similar studies performed worldwide.

The most important conclusion from their study is that even though genetics and genomics is taught in most nursing programs, there is a lack of clinical perspective which makes the nurses incapable of offering genomics-based advice to the patients. The authors further suggest ways to improve the various shortcomings identified in their study. Finally, the authors also acknowledge the limitations of their study.

General Comments: Overall, this article is well written and provides a valuable resource for readers and course coordinators to effectively include the study of genetics and genomics in nursing programs.

1)     Can the authors comment on the institutions of the 138 course coordinators and 49 instructors who participated in the study? That is, are some of them from the same institute? And does that help derive any correlation?

2)     The authors acknowledge that one of the limitations of their study is the imbalance in group size of the instructors (49) and course coordinators (138). Can they further comment if this imbalance might have resulted in the difference in competencies observed (especially C23)? Can the difference be accounted for in the statistical analyses performed in their study to get more accurate correlations?

Author Response

Comments and Suggestions for Authors

In this article, the authors investigate how genetics and genomics are taught in the undergraduate nursing programs in Brazil. They have developed a thorough questionnaire and include both instructors and coordinators of the courses from higher education institutions in Brazil to fill out the survey. They use apt statistical methods to analyze and deduce correlations. The authors also compare their study and conclusions with other similar studies performed worldwide. The most important conclusion from their study is that even though genetics and genomics is taught in most nursing programs, there is a lack of clinical perspective which makes the nurses incapable of offering genomics-based advice to the patients. The authors further suggest ways to improve the various shortcomings identified in their study. Finally, the authors also acknowledge the limitations of their study.

General Comments: Overall, this article is well written and provides a valuable resource for readers and course coordinators to effectively include the study of genetics and genomics in nursing programs.

Response: Thank you very much for your positive feedback.

  • Can the authors comment on the institutions of the 138 course coordinators and 49 instructors who participated in the study? That is, are some of them from the same institute? And does that help derive any correlation?

Response: Thanks for this important comment. In that sense, we've added this in the methods (data collection) section to make it more clear to readers.

In fact, some of the instructors who teach genetics/genomics were from the same institution as the coordinators of the respective Undergraduate Nursing Courses.

This is because the data collection flowchart began with the invitation sent to the coordinators of the Undergraduate Nursing Courses in Brazil. The coordinators were the ones who informed us about the contact of the instructors/professors of the institution that teaches the contents of genetics and genomics to nurses. It is noteworthy that five attempts were made to telephone, in addition to e-mails to each professor on different days and at different times (morning, afternoon and evening), to achieve success. Even so, the response rate of this group of participants was low. As a last attempt to increase the number of responses from genetics professors as much as possible, a request was made to support this research from the Brazilian Society of Genetics (SBG) and the Brazilian Society of Medical Genetics (SBGM), which resulted in a few contacts .

In addition, it is important to emphasize that all coordinators of the Undergraduate Course in Nursing are all nurses, whereas the professors who taught genetics/genomics, mainly teach by biologists (77.6%), and only 8.2% were nurses. However, this fact does not help derive any correlation. So, Pearson's correlation matrix in the bivariate analysis would be unfeasible.

  • The authors acknowledge that one of the limitations of their study is the imbalance in group size of the instructors (49) and course coordinators (138). Can they further comment if this imbalance might have resulted in the difference in competencies observed (especially C23)? Can the difference be accounted for in the statistical analyses performed in their study to get more accurate correlations?

Response: Yes, this imbalance may have masked some finding, and, this might have resulted in the difference in competencies observed. In this sense, we have added this issue in the limitation section of the study. As our sample was small, stratifying it and performing multivariate analyzes would not be adequate. Therefore, we choose not to held any multivariate analysis.

Reviewer 2 Report

1. Genetic and Genomics Teaching in Nursing Programs in a Latin American Country, however, mainly teach by biologists (77.6%), it would be much better to have clinical background! This should be emphasized at discussion in the future how to improve, like in-service education. Fundamental concepts of genetics is limited for clinical situations.

2. Competencies in genetic and genomics that are assessed through certification, need more elaboration.

3. The content of the course should be presented, like the role of genetic nurses and genetic counselors. pedigree taking, ELSI, psycho-social aspect, support group, etc. Students' feedback also need to be included. How to incorporate experiential resources into the courses.

Author Response

Comments and Suggestions for Authors

  1. Genetic and Genomics Teaching in Nursing Programs in a Latin American Country, however, mainly teach by biologists (77.6%), it would be much better to have clinical background! This should be emphasized at discussion in the future how to improve, like in-service education. Fundamental concepts of genetics is limited for clinical situations.

Response: Thank you very much for this valuable and timely comment. We definitely agree with you. We have expanded on these aspects in the Discussion section as per recommended.

  1. Competencies in genetic and genomics that are assessed through certification, need more elaboration.

Response: We have added two paragraphs in this regard as per your suggestion. Thank you!

  1. The content of the course should be presented, like the role of genetic nurses and genetic counselors. pedigree taking, ELSI, psycho-social aspect, support group, etc. Students' feedback also need to be included. How to incorporate experiential resources into the courses.

Response: Thank you very much for this valuable and relevant comment. This point is indeed crucial in the training of nurses. In this sense, we have incorporated some of your kind suggestions at the end of the paper.

Yours Sincerely,

The authors